# A Linear Regression and Deep Learning Approach for Detecting Reliable Genetic Alterations in Cancer Using DNA Methylation and Gene Expression Data

**DOI:** 10.3390/genes11080931

**Published:** 2020-08-12

**Authors:** Saurav Mallik, Soumita Seth, Tapas Bhadra, Zhongming Zhao

**Affiliations:** 1Center for Precision Health, School of Biomedical Informatics, The University of Texas Health Science Center at Houston, Houston, TX 77030, USA; saurav.mallik@uth.tmc.edu or; 2Department of Computer Science & Engineering, Aliah University, Newtown WB-700160, India; soumita.seth@gmail.com (S.S.), tapas.bhadra@gmail.com (T.B.); 3Human Genetics Center, School of Public Health, The University of Texas Health Science Center at Houston, Houston, TX 77030, USA; 4MD Anderson Cancer Center UTHealth Graduate School of Biomedical Sciences, Houston, TX 77030, USA

**Keywords:** uterine cervical cancer, DNA methylation, Liner regression, deep learning, differentially expressed genes

## Abstract

DNA methylation change has been useful for cancer biomarker discovery, classification, and potential treatment development. So far, existing methods use either differentially methylated CpG sites or combined CpG sites, namely differentially methylated regions, that can be mapped to genes. However, such methylation signal mapping has limitations. To address these limitations, in this study, we introduced a combinatorial framework using linear regression, differential expression, deep learning method for accurate biological interpretation of DNA methylation through integrating DNA methylation data and corresponding TCGA gene expression data. We demonstrated it for uterine cervical cancer. First, we pre-filtered outliers from the data set and then determined the predicted gene expression value from the pre-filtered methylation data through linear regression. We identified differentially expressed genes (DEGs) by Empirical Bayes test using Limma. Then we applied a deep learning method, “*nnet*” to classify the cervical cancer label of those DEGs to determine all classification metrics including accuracy and area under curve (AUC) through 10-fold cross validation. We applied our approach to uterine cervical cancer DNA methylation dataset (NCBI accession ID: GSE30760, 27,578 features covering 63 tumor and 152 matched normal samples). After linear regression and differential expression analysis, we obtained 6287 DEGs with false discovery rate (FDR) <0.001. After performing deep learning analysis, we obtained average classification accuracy 90.69% (±1.97%) of the uterine cervical cancerous labels. This performance is better than that of other peer methods. We performed in-degree and out-degree hub gene network analysis using Cytoscape. We reported five top in-degree genes (PAIP2, GRWD1, VPS4B, CRADD and LLPH) and five top out-degree genes (MRPL35, FAM177A1, STAT4, ASPSCR1 and FABP7). After that, we performed KEGG pathway and Gene Ontology enrichment analysis of DEGs using tool *WebGestalt(WEB-based Gene SeT AnaLysis Toolkit)*. In summary, our proposed framework that integrated linear regression, differential expression, deep learning provides a robust approach to better interpret DNA methylation analysis and gene expression data in disease study.

## 1. Introduction

DNA methylation has been found a promising biomarker in cancer detection and cancer classification. DNA methylation can be defined as a heritable epigenetic mark where a methyl group can transfer covalently to the C-5 position of the cytosine ring of DNA through DNA methyltransferases (DNMTs). DNA methylation is vital for normal development. It plays very important role in a number of key operations including genomic imprinting, inactivation of X-chromosome, repression of repetitive element transcription and transposition, and different diseases including cancer [1]. To biologically interpret the DNA methylation data, two kinds of analysis are available: (i) single differentially methylated genes (CpG sites) finding [2,3] and (ii) differentially methylated region (DMR) finding [4,5,6]. These two kinds of analysis are only specific to performing a single task. Therefore, it is important to incorporate different factors to correctly interpret DNA methylation data by which it can work as multi-functionalities from different directions such as prediction of gene expression using DNA methylation, differential expression analysis, cancer classification [7], hub gene finding, and others.

In practical scenarios, it is observed that DNA methylation normally reduces gene expression levels [8,9]. However, this opinion varies on different factors. There are different kinds of method to integrate DNA methylation and gene expression data. There are several shortcomings of those existing methods. Firstly, it is not easy to determine the directionality of the evaluated gene expression estimated from the DNA methylation. Normally, the suppression of gene expression is caused by hypermethylation in the promoter region, while the activation correlates the hypermethylation in the gene body. Therefore, the prediction of changing in gene expression based on simple DNA methylation results is difficult [10]. Secondly, an accurate measure of gene promoter methylation is difficult due to the variance in the size of canonical promoters as well as the presence of the distal augments, which initiates biases into the association of methylated regions with gene models [10]. Thirdly, the high probability of selecting a long gene due to the nearby differentially methylated CpGs or overlapping (or nested) with other genes [10]. Fourthly, some specific tools are required for reformatting the methylation data into the genomic region formats (e.g., BED) for some web-based methods such as GREAT [11], Galaxy [12]. It creates more complications in their usage [10].

Cervical cancer is a cancer which starts in the cervix, a hollow cylinder that connects the lower part of uterus to a woman’s vagina. Most of the cervical cancers grow in the cells on the outer surface of the cervix. Normally women are unable to realize this disease in the initial stage since the symptoms are more or less similar with the common conditions such as menstrual periods and urinary tract infections. The normal symptoms of the cervical cancer include abnormal bleeding during mensuration time or after having sex, pain in the pelvis, as well as pain during the urination [13]. Here, we used a DNA methylation dataset for uterine cervical cancer from NCBI (Accession ID: GSE30760) [14] which have two types of samples, one is normal sample and another one is uterine cervical cancer sample.

So far, there has been no method to integrate regression, differential expression and deep learning strategies for accurate interpretation of DNA methylation in a complex disease like cancer. To resolve the previously mentioned drawbacks, in this article, we provided an integrated framework using regression, differential expression and deep learning methods to correctly interpret biologically of the DNA methylation data through integrating that DNA methylation data and corresponding TCGA (The Cancer Genome Atlas) gene expression data for uterine cervical data (NCBI accession ID GSE30760) [14,15,16]. We pre-filtered the redundant CpG sites, eliminated outliers, and replaced missing values. Next, we predicted corresponding gene expression value from the pre-filtered DNA methylation data through linear regression algorithm where the impact between DNA methylation and TCGA gene expression has been determined. As a result, we obtained the predicted gene expression matrix for the preprocessed DNA methylation data. Through the entire analysis, we used ByMethyl R package [10]. Next, we identified differentially expressed genes (DEGs) using downstream analysis, Empirical Bayes test using Limma [17,18,19]. After we applied a recently released deep learning method, “*nnet*” (feed-forward neural network based model) [20] to interpret those DEGs for determining the classification capacity of uterine cancer and normal groups, we then estimated all classification metrics (average accuracy, average sensitivity, average specificity, average precision, average overall error rate and area under curve (AUC)) using 10-fold cross validation. We trained our predicted DEG expression data using “*nnet*” with the default parameter settings (i) size (=number of units in hidden layer), (ii) rang (=initial random weights) while [−rang, rang], (iii) decay (=parameter for weight decay), (iv) maxit (=the maximum number of iterations or number of epochs), (v) MaxNWts (=the maximum allowable number of weights) and other default parameters. Remarkably, we obtained 90.69% (±1.97%) as average classification accuracy of the uterine cervical cancer samples and normal samples by using DEG expression data. According to comparative study, the classification accuracy of our proposed method is higher than that of other state-of-the-art methods. We further performed in-degree and out-degree hub gene network analysis using Cytoscape [21]. We reported the five top in-degree genes (PAIP2, GRWD1, VPS4B, CRADD and LLPH) and the five top out-degree genes (MRPL35, FAM177A1, STAT4, ASPSCR1 and FABP7). After that, we performed Gene Set Enrichment Analysis (GESA) to determine enriched KEGG pathways and Gene Ontology (GO) terms including Biological Process (BP), Cellular Component (CC), and Molecular Function (MF) on the set of all DEGs having FDR<0.001 using *WebGestalt (WEB-based Gene SeT AnaLysis Toolkit)* [22]. Finally, our proposed integrated framework using linear regression, differential expression and deep learning method can interpret the DNA methylation data better than using single differential methylation analysis or differentially methylated region finding strategies for any kind of cancer.

## 2. Materials and Methods

The steps of our proposed framework are demonstrated as follows, as well as in Figure 1.

### 2.1. Data Collection

In this study, we used a cervical cancer methylation dataset(NCBI accession ID: GSE30760) [14,15,16]. This dataset included 63 uterine cervical tumor samples and 152 matched normal samples. Of note, the initial analysis had 27,578 genes.

### 2.2. Preprocessing of Methylation Data

In this article, we provided an extensive analysis to integrate DNA methylation and corresponding TCGA gene expression data by utilization of regression, differential expression and deep learning. In this method, we have utilized different steps as below.

#### 2.2.1. Data Preprocessing

First we eliminated the CpG sites that had missing values in more than half of the samples and then the remaining missing values would be imputed through integrating a new traditional quality control R package $‘{\mathrm{ENmix}}^{\prime }$ [23], which is widely useful in Illumina Human Methylation data analysis. The functions under the $‘{\mathrm{ENmix}}^{\prime }$ package are used to remove unwanted experimental noise and to improve accuracy and reproducibility of methylation measures. ENmix functions are very flexible and transparent. In our proposed method this quality control ‘*ENmix*’ was used in our methylation data to discard the outliers and to replace missing values using the popular k nearest neighbors (KNN) algorithm.

#### 2.2.2. Computing Predicted Expression Scores of Gene through Linear Regression Analysis

In this step, we computed the predicted gene expression scores from the preprocessed of DNA methylation profiles and corresponding TCGA CESC cancer type through linear regression analysis along with corresponding pre-trained weight factor.

To do so, we utilized the linear regression algorithm to measure the impact between DNA methylation and gene expression for uterine cervical cancer on preprocessed DNA methylation and corresponding TCGA CESC cancer type [10]. In a statistical point of view, linear regression is a linear approach for molding the relationship between a scalar variable (or, dependent variable) and one or more explanatory variables (or independent variables). In regression analysis, gene expression ($E_{x}$) is the dependent variable and DNA methylation (Mt) is the independent variable. For an i-indexed gene denoted by $gene_{i}$, $E_{x}=\left\{e_{x_{1}},e_{x_{2}},...,e_{x_{n}}\right\}$ is the gene expression across *n* samples, and Mt is the corresponding methylation matrix (27,578×215 matrix here). Here, we chose those CpGs ($cpg_{n,j}$) whose beta values were correlated, i.e., Pearson’s correlation coefficient was greater than |0.8|) with gene expression label ($gene_{i}$) for building the model, $gene_{i}$ where $cpg_{n,j}$ is the beta value of j-th CpG in sample *n*. The equation for the linear regression model was described as follows:(1)$$E_{x}=\alpha +beta*Mt$$
where α denotes the linear regression intercepting factor, and beta refers to the coefficient vector. In our case, through this linear regression model, we generated the predicted gene expression matrix for the provided genes (CpG sites) using DNA methylation data. Then we applied 10-fold cross-validation to validate our model. That means, we need to check the quality of the gene expression inferred by the linear regression model. Basically, for each validation, to train the model we used 9/10 samples as training dataset. Then, we computed a gene expression profile for the rest 1/10 samples by integrating the DNA methylation data and trained model. After completion of 10-fold cross-validation, our further step was to merge test sample profiles to a gene expression profile containing all samples. For conducting downstream validation we compared the gene expression with the RNA-seq data.

#### 2.2.3. Voom Normalization and Identifying Differentially Expressed Genes Using Limma

In this step Voom normalization [24] was used and after that we applied Limma [18,25]. After applying Voom normalization tool, we detected DEGs from the predicted gene expression data for downstream analysis through Limma [19]. According to benchmark methods the performance of Limma is very good for any kind of data distributions for any sample size. The definition of the moderated *t*-statistic of Limma is as follows [19]:(2)$${\bar{t}}_{k}=\left(\frac{1}{\sqrt{\left(\frac{1}{m_{1}}\right)+\left(\frac{1}{m_{2}}\right)}}\right)*\left(\frac{\hat{\beta_{k}}}{{\bar{p}}_{k}}\right)$$
where $m_{1}$ denotes the sample size for diseased group and $m_{2}$ signifies the sample size for control group, and total sample size $m=m_{1}+m_{2}$. $\hat{\beta_{k}}$, $p_{k}$ notify corresponding contrast estimator and posterior sample variance for the genes, respectively.

To find the false discovery rate (FDR) adjusted *p*-value using Empirical Bayes *t*-statistics, we used *t*-table or cumulative distribution function (cdf). FDR adjusted *p*-value less than 0.001 indicates the differentially expressed genes (DEGs) here. This *p*-value denotes the probability of observing a *t*-value which is either equal to or greater than the actually observed *t*-value in which the given null hypothesis is true.

Here, we applied the Empirical Bayes test using Limma to compute *t*-score, *p*-value and FDR, where normal uterine samples group had 152 samples and uterine cervical cancer samples group included 63 samples. Finally, we selected those genes as differentially expressed genes whose FDR<0.001. However, all the differentially expressed genes were considered as a single potential gene signature which could be verified at classification analysis through deep learning.

### 2.3. Disease Classification of DEGs through Deep Learning

Here, we used a latest deep learning method “*nnet*” (feed-forward neural network based model), [20]. We used this deep learning technique with 10-fold cross validation to examine the class-label (normal and Uterine Cervical cancer groups) of the differentially expressed genes with a repeat of thirty times. In the cross-validation, we divided the predicted gene expression data of the DEGs into 10 folds of samples of which nine folds of samples were used as training set, while remaining one fold of samples was utilized as the test set. From this sub step, we ran “*nnet*” tool using a certain number of epochs (termed as “maxit”) that means the deep learning method was internally repeated for that “maxit” times, and then computed the classification metrics at one time iteration of each fold. From this sub step, we obtained a confusion matrix consisting of True Positive (TP), False Negative (FN), False Positive (FP) and True Negative (TN). This sub procedure was repeated for each fold of samples (i.e., nine other fold samples). Then we added all these metrics for these 10 times internal repetition and then produced a final confusion matrix. Then we added all these metrics for these 10 internal repetitions and then produced a final confusion matrix. Thereafter, we repeated this entire procedure multiple times (30 times) here to obtain the average classification metric values (average accuracy, average sensitivity, average specificity, average precision, average overall error rate and area under curve (AUC)). Here, we used the test sample as a validation sample also. In this deep learning method, we used “*nnet*” with the default parameter settings (i) size (=number of units in hidden layer), (ii) rang (=initial random weights) while [−rang, rang], (iii) decay (=parameter for weight decay), (iv) maxit (=the maximum number of iterations or number of epochs), (v) MaxNWts (=the maximum allowable number of weights) and other default parameters also.

### 2.4. Hub Gene Finding

In this regard, we applied Pearson’s correlation analysis on the DEGs identified by our method for finding out the active edges among genes having correlation value ≥0.8 or ≤−0.8. After obtaining the set of active edges, we performed degree centrality analysis through Cytoscape online tool [21] and determined in-degree and out-degree scores of each DEG. We marked top 10 in-degree hub DEGs and top 10 out-degree hub DEGs.

### 2.5. Gene Set Enrichment Analysis

The potential function, biological significance, and disease relevance of a list of signature genes can be assessed by Gene Set Enrichment Analysis (GSEA). After identifying differentially expressed genes we used KEGG pathways and Gene Ontology (GO) annotations (three domains: Biological Process (BP), Cellular Component (CC), and Molecular Function (MF)) on a set of top differentially expressed genes by *WebGestalt (WEB-based Gene SeT AnaLysis Toolkit)* [22]. We obtained all KEGG pathways and Gene Ontology (GO) terms accompanied by number of genes in that pathway or GO-term, enriched *p*-value and FDR. We filtered out those KEGG pathways or GO terms whose FDR was greater than or equal to 0.05.

## 3. Results and Discussion

In this case study, we had 27,578 features and 215 samples including 152 normal samples and 63 uterine cervical cancer samples. After data preprocessing, linear regression and differential expression analysis, we obtained 6287 DEGs having FDR<0.001 by Limma, in a list accompanied by computed *t*-score, *p*-value and FDR. Top 25 DEGs are shown in Table 1. For example, ADCY2 was the topmost DEG with minimum FDR (FDR = $5.64\times 10^{-119}$). We provided the list of all DEGs obtained by differential expression analysis by Empirical Bayes test using Limma with FDR corrected *p*-value in a Appendix A, Additional file 1: Appendix A. Furthermore, the predicted gene expression matrix of all DEGs computed from original pre-filtered uterine cervical cancer DNA methylation data through linear regression analysis was provided in another Appendix A, Additional file 2: Appendix A.

After that, we applied the latest deep learning method “*nnet*” (feed-forward neural network based model), [20] on our computed DEG expression dataset which have 6287 features with 215 samples. We used this deep learning technique with 10-fold cross validation to examine the class-label (normal and uterine cervical cancer groups) of the differentially expressed genes with a repeat of 30 times. In the cross-validation, we divided all the samples of the predicted gene expression data of the DEGs into 10 folds of samples of which nine-fold of samples was used as training set, while the remaining one-fold of the samples was utilized as the test set. From this sub step, we ran “*nnet*” tool using maxit (number of epochs) equal to 100, that means the deep learning method was internally repeated for 100 times, and then computed the classification metrics at one time iteration of each fold. From this sub step, we obtained a confusion matrix consisting of True Positive (TP), False Negative (FN), False Positive (FP) and True Negative (TN). This sub procedure was repeated for each fold of samples (i.e., nine other folds). Then, we added all these metrics for these 10 times internal repetitions and produced a final confusion matrix. Thereafter, we repeated this entire procedure for multiple times (30 times) and obtained thirty confusion metrics. Using this, we obtained the average classification metric values (average accuracy, average sensitivity, average specificity, average precision, average overall error rate and area under curve (AUC)). Note that our deep learning method has already repeated 30,000 times (30×10×100) from which we computed the average accuracy, where every sample was used as a test set at least once (i.e., no sample was missing as a test sample). Here we used test sample as validation 163 sample. In this deep learning method, we used “*nnet*” with the default parameter settings (i) size (=number of units in hidden layer) (=2), (ii) rang (=initial random weights)(=0.1) while [−rang, rang], (iii) decay (=parameter for weight decay)(=$5\times e^{-4}$), (iv) maxit (=the maximum number of iterations or number of epochs)(=100), (v) MaxNWts (=the maximum allowable number of weights)(=84,581) and other default parameters. As we used 10-fold cross validation, 9/10 of 215 samples (i.e., 194 or 193 samples) were considered as training set and 1/10 of 215 samples (i.e., 21 or 22 samples) were taken as test set. of which nine-fold of samples was used as a training set, while remaining one-fold of samples was utilized as a test set. Thus, each sample participated in each role, either in training sample or test sample, at least once. Here, we also used the test sample as the validation sample. We obtained 90.69% (±1.97%) average classification accuracy and value of AUC was 0.858. For more details, see Table 2. We have plotted all metrics in Figure 2.

We carried out a comparative study between proposed method and an existing method “*RSNNS*” (Stuttgart Neural Network Simulator (SNNS) based deep learning tool) with 10-fold cross validation with repeating 30 times. In case of “*RSNNS*” we also used same default parameter settings like (i) size (=number of units in hidden layer)(=2), (ii) maxit (=maximum number of iterations or number of epochs) (=100), among others. In both cases we have repeated entire procedure 30 times to to obtain a reliable classification. Our proposed method produced an average classification accuracy of 90.69% (±1.97%) whereas existing method “*RSNNS*” had 87.27% (±5.92%) as average classification accuracy (see Figure 3). We considered our framework had better performance than all other methods using deep learning tool.

Here, we applied Pearson’s correlation analysis on our DEGs for finding out edges among genes having correlation value greater than or equal to 0.8 or, less than or equal to (−0.8). Then, we also performed in-degree and out-degree hub gene network analysis using Cytoscape [21]. As an example the five top genes with highest in-degree values were namely PAIP2, GRWD1, VPS4B, CRADD and LLPH, see Table 3. Similarly, the five top most out-degree genes were namely MRPL35, FAM177A1, STAT4, ASPSCR1 and FABP7, see Table 4. We provided detail hub gene network structure in a Appendix A, Additional file 7: Appendix A.

In the corresponding literature survey, we found that most of the topmost hub genes detected by our method played an important role in the respective cancer. PAIP2 gene and cervical cancer were found to be associated by Berlanga et al. [26]. GRWD1 was utilized as the negatively regulator of p53 in tumorigenesis [27]. It had been also used as a potential bio-marker in DNA methylation at the time of treatment and risk assessment of cancer. Methylation of GRWD1 might be a protective factor in the development of tumor [28]. VPS4B gene and cervical cancer were reported in the literature Broniarczyk et al. [29]. Similarly, CRADD gene is involved in cervical cancer, as reported in Sundaram et al. [30], while LLPH gene was associated with cervical cancer in Feron et al. [31]. Similarly, the topmost out-degree hub genes were mostly associated with cervical cancer through literature search. For example, the association between FAM177A1 and cervical cancer were documented in Wen et al. [32], whereas STAT4 was connected with the respective cervical cancer in Luo et al. [33]. In addition, ASPSCR1 and cervical cancer are reported in Liang et al. [34], while FABP7 was found to be linked to cervical cancer in Zhang et al. [35].

These 6287 DEGs, which have FDR<0.001, were taken for Gene Set Enrichment Analysis using *WebGestalt (WEB-based Gene SeT AnaLysis Toolkit)* [22]. We had applied *WebGestalt* (WEB-based Gene SeT AnaLysis Toolkit) database on our DEG set to obtain all KEGG pathways and Gene Ontology (GO) terms [Biological Process (BP), Cellular Component (CC) and Molecular Function (MF)], accompanied by number of genes in that pathway or GO-term, enriched *p*-value and FDR. Here, we took our input data set in the prescribed format of WebGestalt which was in a two-columns pattern, first one was gene name and second one was score. Here we used *t*-score as score. Significant pathways and GO-terms were described in below and also for more details see Table 5, Table 6, Table 7 and Table 8. For example, *hsa05205:Proteoglycans in cancer* was a top significant KEGG pathway which has minimum FDR value ($2.16\times 10^{-5}$). A total of 198 genes were associated in this pathway with enriched *p*-value $6.65\times 10^{-8}$. For the remaining top 10 significant KEGG pathways, see Table 5. We provided the list of all KEGG pathways in a Appendix A, Additional file 3: Appendix A. In addition, the volcano plot of the of normalized enrichment score of those FDR significant KEGG pathways is shown in Figure 4. Similarly, *GO:0008283 cell proliferation* was one of the top significant GO-BP terms with FDR value 0. A total of 1986 genes were associated with this GO-BP term, enriched p-value 0. For the remaining terms, see Table 6. We provided the list of all GO-BP terms in a Appendix A, Additional file 4: Appendix A. In such analysis, we found *GO:0005783 endoplasmic reticulum* as one of the top significant GO-CC terms with FDR value 0. A total of 1861 genes were associated with this GO-CC term, enriched *p*-value 0. For the rest, see Table 7. We provided the list of all GO-CC terms in a Appendix A, Additional file 5: Appendix A. Furthermore, *GO:0042802 identical protein binding* was one of the top significant GO-MF terms with minimum FDR value 0. A total of 1696 genes were associated with this GO-MF term having the enriched *p*-value 0. For details, see Table 8. We provided the list of all GO-MF terms in a Appendix A, Additional file 6: Appendix A.

## 4. Conclusions and Future Work

In this article, we provided a framework using linear regression, differential expression, and deep learning to provide correct biological interface for integrating DNA methylation and corresponding TCGA gene expression data to uterine cervical cancer. To develop the framework, first we eliminated outliers, then applied linear regression to determine predicted gene expression data from the preprocessed DNA methylation data by the use of TCGA gene expression data. Then we identified 6287 differentially expressed gene with FDR cut off less than 0.001 using downstream analysis through Empirical Bayes test using Limma. After that, we applied “*nnet*” deep learning method to interpret differentially expressed genes with 10-fold cross validation and with the default parameter settings (i) size (=number of units in hidden layer), (ii) rang (=initial random weights) while [−rang, rang], (iii) decay (=parameter for weight decay), (iv) maxit (=the maximum number of iterations or number of epochs), (v) MaxNWts (=the maximum allowable number of weights) and other default parameters also. We obtained 90.69% (±1.97%) as average classification accuracy of the uterine cervical cancer samples and normal samples for DEG expression data, which is more significant than other existing methods. So through the deep learning and comparative study, we can say that our obtained DEGs are strong and efficient in disease classification.

Here, we also performed in-degree and out-degree hub gene network analysis using Cytoscape [21]. We reported the five highest in-degree genes (PAIP2, GRWD1, VPS4B, CRADD and LLPH) and the five highest out-degree genes (MRPL35, FAM177A1, STAT4, ASPSCR1 and FABP7). Furthermore, we used pathway analysis on DEGs with FDR<0.001 using WebGestalt. Finally, our framework is useful for better biological interpretation of the DNA methylation data rather than single differential methylation analysis or differentially methylated region finding.

In our future study, we will extend our current work through integrating random forest ensemble method into deep learning strategy to obtain a better classification model in all prospective, and then apply that on big data (e.g., single cell RNA sequencing data or, other TCGA cancer tissue specific data) for cancer classification.

## Figures and Tables

**Figure 1 genes-11-00931-f001:**
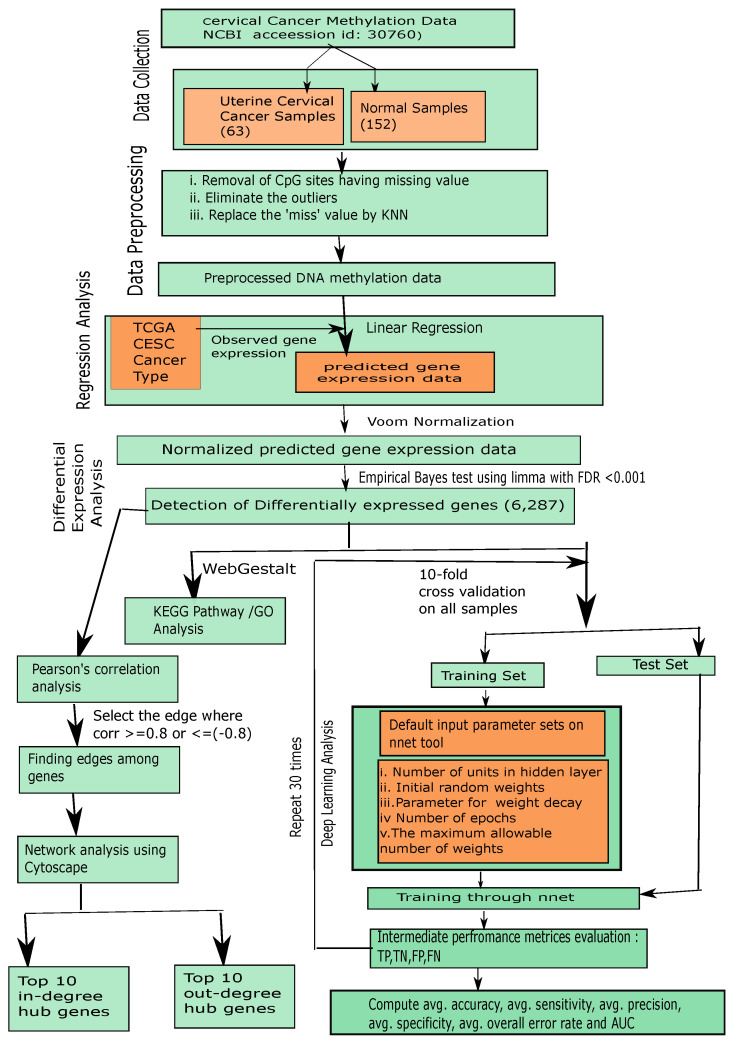
Flowchart of the proposed framework.

**Figure 2 genes-11-00931-f002:**
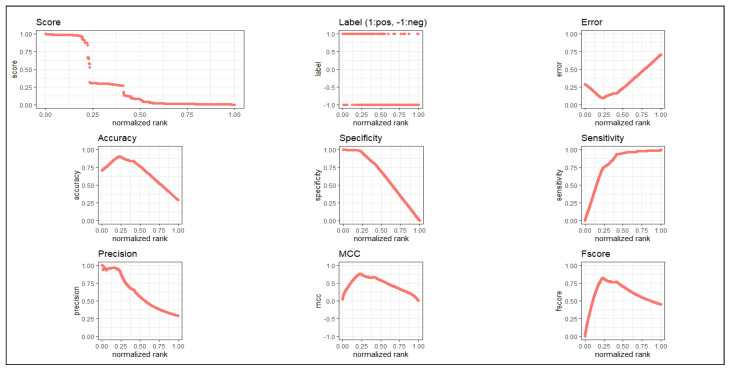
ROC plots of all classification metrics for the proposed method.

**Figure 3 genes-11-00931-f003:**
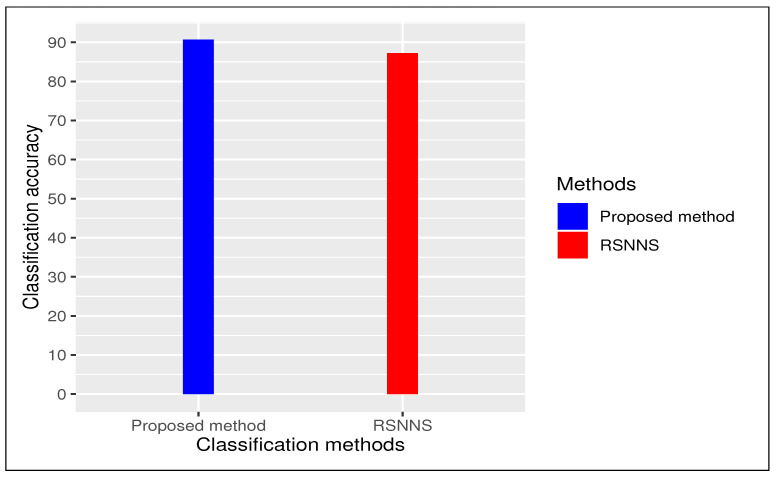
Comparative bar plot: proposed method vs state-of-the-art method (RSNNS)).

**Figure 4 genes-11-00931-f004:**
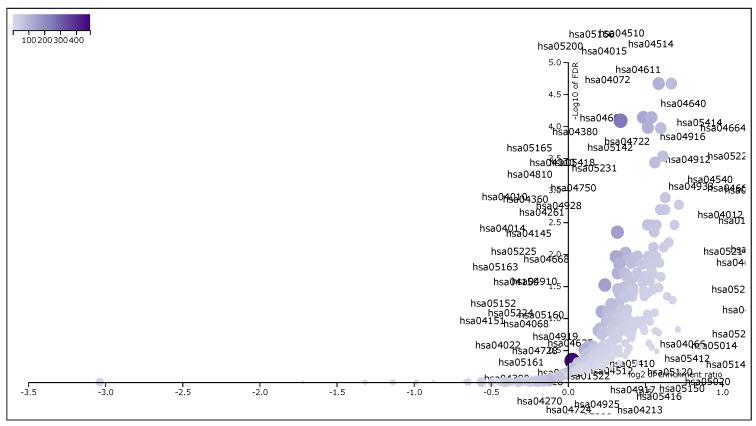
The volcano plot of normalized enrichment score of the FDR significant KEGG pathways from GSEA analysis of DEGs.

**Table 1 genes-11-00931-t001:** List of differentially expressed genes (false discovery rate (FDR) sorted).

Gene Symbol	*t*-Score	*p*-Value	FDR
ADCY2	45.22	$5.64\times 10^{-119}$	$5.95\times 10^{-115}$
PTPN6	32.50	$1.43\times 10^{-89}$	$7.55\times 10^{-86}$
LHFPL2	32.09	$1.63\times 10^{-88}$	$5.71\times 10^{-85}$
VAV1	30.24	$1.41\times 10^{-83}$	$3.72\times 10^{-80}$
EYA4	−29.40	$2.97\times 10^{-81}$	$6.27\times 10^{-78}$
PNPLA2	29.02	$3.38\times 10^{-80}$	$5.94\times 10^{-77}$
ARID3A	−28.71	$2.37\times 10^{-79}$	$3.56\times 10^{-76}$
HOXD10	28.19	$6.97\times 10^{-78}$	$9.20\times 10^{-75}$
TWIST1	27.69	$1.85\times 10^{-76}$	$2.17\times 10^{-73}$
BHMT	26.49	$5.42\times 10^{-73}$	$5.72\times 10^{-70}$
TSLP	26.25	$2.76\times 10^{-72}$	$2.65\times 10^{-69}$
ACCN4	26.16	$5.23\times 10^{-72}$	$4.60\times 10^{-69}$
HOXA6	25.94	$2.22\times 10^{-71}$	$1.80\times 10^{-68}$
PRR5	25.67	$1.40\times 10^{-70}$	$1.06\times 10^{-67}$
NODAL	25.45	$6.44\times 10^{-70}$	$4.53\times 10^{-67}$
EFCAB1	25.41	$8.60\times 10^{-70}$	$5.45\times 10^{-67}$
WNT2	25.40	$8.86\times 10^{-70}$	$5.45\times 10^{-67}$
PC	25.40	$9.31\times 10^{-70}$	$5.45\times 10^{-67}$
S100A8	25.23	$2.84\times 10^{-69}$	$1.58\times 10^{-66}$
VWCE	24.89	$3.05\times 10^{-68}$	$1.61\times 10^{-65}$
IGFBP2	24.86	$3.61\times 10^{-68}$	$1.81\times 10^{-65}$
ZNF385A	24.74	$8.36\times 10^{-68}$	$4.01\times 10^{-65}$
C1orf220	24.71	$1.04\times 10^{-67}$	$4.75\times 10^{-65}$
COG2	24.63	$1.85\times 10^{-67}$	$8.15\times 10^{-65}$
QRFP	−24.51	$4.16\times 10^{-67}$	$1.76\times 10^{-64}$

**Table 2 genes-11-00931-t002:** Values of disease classification metrics by proposed method.

Metrics	Average Value (std *)
Average accuracy	90.69% (±1.97%)
Average sensitivity	73.97% (±1.06%)
Average specificity	97.63% (±1.71%)
Average precision	93.38% (±4.17%)
Average overall error rate	9.30% (±1.97%)
Area under curve (AUC)	0.858

* std: standard deviation.

**Table 3 genes-11-00931-t003:** Top 10 hub genes according to the in-degree centrality from our proposed method.

Gene Symbol	In-Degree	Out-Degree	Average Shortest Path Length	Betweenness Centrality	Closeness Centrality	Clustering Coefficient
PAIP2	439	32	3.587	0.802	0.279	0.188
GRWD1	425	66	3.435	11.001	0.291	0.178
VPS4B	406	68	3.460	2.276	0.289	0.191
CRADD	406	178	3.087	11.003	0.324	0.152
LLPH	403	40	3.545	2.313	0.282	0.182
NDUFA4	390	89	3.556	1.927	0.281	0.168
NDUFB6	372	111	3.294	4.661	0.304	0.175
ZKSCAN4	372	88	3.364	1.434	0.297	0.200
SMARCD1	365	43	3.515	0.734	0.284	0.214
TMED10	348	39	3.546	4.124	0.282	0.193

**Table 4 genes-11-00931-t004:** Top 10 hub genes according to the out-degree centrality from our proposed method.

Gene Symbol	In-Degree	Out-Degree	Average Shortest Path Length	Betweenness Centrality	Closeness Centrality	Clustering Coefficient
MRPL35	239	376	2.765	9.354	0.362	0.141
FAM177A1	21	339	3.002	0.263	0.333	0.225
STAT4	94	332	2.872	2.744	0.348	0.211
ASPSCR1	68	329	2.888	1.132	0.346	0.212
FABP7	204	315	2.779	3.008	0.360	0.171
HNRNPA0	65	311	3.010	1.230	0.332	0.191
ANGPTL4	18	299	2.887	0.364	0.346	0.249
DDX19A	86	283	2.993	1.385	0.334	0.218
TRNT1	40	282	3.119	0.477	0.321	0.221
PFDN1	52	274	3.005	0.526	0.333	0.243

**Table 5 genes-11-00931-t005:** Top significant KEGG Pathways (FDR sorted).

KEGG Pathway Name *	#Genes	Enriched *p*-Value	FDR
*hsa05205 Proteoglycans in cancer*	198	$6.65\times 10^{-8}$	$2.16\times 10^{-5}$
*hsa04550 Signaling pathways regulating pluripotency of stem cells*	139	$1.32\times 10^{-7}$	$2.16\times 10^{-5}$
*hsa05166 Human T-cell leukemia virus 1 infection*	255	$6.95\times 10^{-7}$	$7.29\times 10^{-5}$
*hsa04510 Focal adhesion*	199	$8.94\times 10^{-7}$	$7.29\times 10^{-5}$
*hsa05200 Pathways in cancer*	524	$1.26\times 10^{-6}$	$8.19\times 10^{-5}$
*hsa04015 Rap1 signaling pathway*	206	$1.93\times 10^{-6}$	$1.05\times 10^{-4}$
*hsa04514 Cell adhesion molecules (CAMs)*	144	$2.31\times 10^{-7}$	$1.07\times 10^{-4}$
*hsa04611 Platelet activation*	123	$7.22\times 10^{-6}$	$2.94\times 10^{-4}$
*hsa04072 Phospholipase D signaling pathway*	146	$1.02\times 10^{-5}$	$3.69\times 10^{-4}$
*hsa04640 Hematopoietic cell lineage*	97	$4.01\times 10^{-5}$	$1.31\times 10^{-3}$

* See Appendix A for details.

**Table 6 genes-11-00931-t006:** Top significant GO-BP term enriched (FDR sorted).

GO-BP Term Name *	#Genes	Enriched *p*-Value	FDR
*GO:0008283 cell proliferation*	1986	0	0
*GO:0006928 movement of cell or subcellular component*	1967	0	0
*GO:0009891 positive regulation of biosynthetic process*	1949	0	0
*GO:0016192 vesicle-mediated transport*	1942	0	0
*GO:0006955 immune response*	1919	0	0
*GO:0031328 positive regulation of cellular biosynthetic process*	1919	0	0
*GO:0006915 apoptotic process*	1911	0	0
*GO:0010628 positive regulation of gene expression*	1911	0	0
*GO:2000026 regulation of multicellular organismal development*	1908	0	0
*GO:0006468 protein phosphorylation*	1860	0	0

* See Appendix A for details.

**Table 7 genes-11-00931-t007:** Top significant GO-CC term enriched (FDR sorted).

GO-CC Term Name *	#Genes	Enriched *p*-Value	FDR
*GO:0005783 endoplasmic reticulum*	1861	0	0
*GO:0097458 neuron part*	1690	0	0
*GO:0031984 intrinsic component of plasma membrane*	1673	0	0
*GO:0031984 organelle subcompartment*	1661	0	0
*GO:0098805 whole membrane*	1630	0	0
*GO:0005887 integral component of plasma membrane*	1596	0	0
*GO:0005794 Golgi apparatus*	1516	0	0
*GO:0044433 cytoplasmic vesicle part*	1462	0	0
*GO:0044463 cell projection part*	1425	0	0
*GO:0120038 plasma membrane bounded cell projection part*	1425	0	0

* See Appendix A for details.

**Table 8 genes-11-00931-t008:** Top significant GO-MF term enriched (FDR sorted).

GO-MF Term Name *	#Genes	Enriched *p*-Value	FDR
*GO:0042802 identical protein binding*	1696	0	0
*GO:0005102 signaling receptor binding*	1538	0	0
*GO:0019904 protein domain specific binding*	684	0	0
*GO:0044212 transcription regulatory region DNA binding*	896	$1.33\times 10^{-15}$	$6.25\times 10^{-13}$
*GO:0001067 regulatory region nucleic acid binding*	898	$2.00\times 10^{-15}$	$7.50\times 10^{-13}$
*GO:0003690 double-stranded DNA binding*	915	$1.02\times 10^{-14}$	$3.16\times 10^{-12}$
*GO:0008134 transcription factor binding*	638	$1.18\times 10^{-14}$	$3.16\times 10^{-12}$
*GO:0016301 kinase activity*	845	$2.19\times 10^{-14}$	$5.00\times 10^{-12}$
*GO:1990837 sequence-specific double-stranded DNA binding*	823	$2.49\times 10^{-14}$	$5.00\times 10^{-12}$
*GO:0000976 transcription regulatory region sequence-specific DNA binding*	781	$2.66\times 10^{-14}$	$5.00\times 10^{-12}$

* See Appendix A for details.

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
