# Peer review of "A Linear Regression and Deep Learning Approach for Detecting Reliable Genetic Alterations in Cancer Using DNA Methylation and Gene Expression Data"

_genes, 2020, doi:10.3390/genes11080931_

Round 1

Reviewer 1 Report

In the current paper, Mallik et al. developed a novel framework to deconvolute the interplay between DNA methylation and gene expression. This is a challenging area for conventional methods due to technical difficulties such as varying promoter sizes. In the current study, they tackled this issue by incorporating a deep learning strategy with conventional regression and differential expression approaches. After extensive cross-validation, the proposed method appeared to outperform the existing RSNNS method. A proof-of-principle analysis was performed in a TCGA cervical cancer dataset, followed by a degree centrality analysis using Cytoscape. Through these analyses, they identified a number of potential hub genes. Several of these genes have already been previously found to be associated with various types of cancer, confirming the validity and potential utility of the pipeline. Overall, the analysis is well-designed and adequately performed, the findings appear to be useful and clinically relevant, and the manuscript is clearly written. One minor issue is it seems there is a lack of discussion about the identified genes. They found five in-degree and five out-degree genes as potential hubs of the regulation. However, there was no further discussion about the function of these genes in cancer. Indeed, several of these genes had already been found to be involved in various tumorigenic processes. For example, GRWD1 had been found to be a negative regulator of p53 in many previous papers. Although it is completely optional, adding these discussions about the functional roles of these genes in cancer may further demonstrate the potential utility of the current tool.

Author Response

Reviewer #1:

Comment:  In the current paper, Mallik et al. developed a novel framework to deconvolute the interplay between DNA methylation and gene expression. This is a challenging area for conventional methods due to technical difficulties such as varying promoter sizes. In the current study, they tackled this issue by incorporating a deep learning strategy with conventional regression and differential expression approaches. After extensive cross-validation, the proposed method appeared to outperform the existing RSNNS method. A proof-of-principle analysis was performed in a TCGA cervical cancer dataset, followed by a degree centrality analysis using Cytoscape. Through these analyses, they identified a number of potential hub genes. Several of these genes have already been previously found to be associated with various types of cancer, confirming the validity and potential utility of the pipeline. Overall, the analysis is well-designed and adequately performed, the findings appear to be useful and clinically relevant, and the manuscript is clearly written.

One minor issue is it seems there is a lack of discussion about the identified genes. They found five in-degree and five out-degree genes as potential hubs of the regulation. However, there was no further discussion about the function of these genes in cancer. Indeed, several of these genes had already been found to be involved in various tumorigenic processes. For example, GRWD1 had been found to be a negative regulator of p53 in many previous papers. Although it is completely optional, adding these discussions about the functional roles of these genes in cancer may further demonstrate the potential utility of the current tool.

Response:  We thank Reviewer #1 for his/her summary of our work and this valuable point.  As per your suggestion, we have added literature survey about the identified hub genes in our manuscript. Using our tool, we detected 5 top genes with highest in-degree values (PAIP2, GRWD1, VPS4B, CRADD and LLPH), as well as the 5 top most out-degree genes (MRPL35, FAM177A1, STAT4, ASPSCR1 and FABP7). 

In the corresponding literature survey, we found that most of the topmost hub genes detected by our method have played an important role in the respective cancer. PAIP2 gene and cervical cancer were found to be associated by Berlanga et al. [23]. GRWD1 was utilized as the negatively regulator of p53 in tumorigenesis [24]. It had been also used as a potential bio-marker in DNA methylation at the time of treatment and risk assessment of cancer. Methylation of GRWD1 might be a protective factor in the development of tumor [25]. VPS4B gene and cervical cancer were reported in Broniarczyk et al. [26]. Similarly, CRADD gene is involved in cervical cancer, as reported in Sundaram et al. [27], while LLPH gene was associated with cervical cancer in Feron et al. [28].

Similarly, the topmost out-degree hub genes were mostly associated with cervical cancer through literature search. For example, the association between FAM177A1 and cervical cancer were documented in Wen et al. [29], whereas STAT4 was connected with the respective cervical cancer in Luo et al. [30]. In addition, ASPSCR1 and cervical cancer are reported in Liang et al. [31], while FABP7 was found to be linked to cervical cancer in Zhang et al. [32].

Please see Section 3 (“Results and Discussion”) on pages 9-10, lines 241-253.

Reviewer 2 Report

Most of the reviewers’ comments were well addressed.

But I am still a bit concerned about the switch from “deepnet” to “nnet”. I know it is a newer R package for neural network. However, the former one is for general deep learning architectures which allows for multiple hidden layers and many other flexible settings. The latter one is just for the simple neural network with only one hidden layer. They are quite different. In addition, the authors only used the default setting of “nnet”. This makes me worry that better classification performance can be easily achieved with more careful tuning of parameters. One innovative component of the paper was on the deep learning. The author responded to many comments by removing it and replacing it with traditional neural network. This is tricky and at the same time a sacrifice on novelty. On a different note, I am almost sure that random forest will perform better than “nnet”. If you want to switch to a method without too many hyper-parameters, why not random forest or gradient boosting tree?

Author Response

Reviewer #2:

Comment:  Most of the reviewers’ comments were well addressed. But I am still a bit concerned about the switch from “deepnet” to “nnet”. I know it is a newer R package for neural network. However, the former one is for general deep learning architectures which allows for multiple hidden layers and many other flexible settings. The latter one is just for the simple neural network with only one hidden layer. They are quite different. In addition, the authors only used the default setting of “nnet”. This makes me worry that better classification performance can be easily achieved with more careful tuning of parameters. One innovative component of the paper was on the deep learning.

The author responded to many comments by removing it and replacing it with traditional neural network. This is tricky and at the same time a sacrifice on novelty. On a different note, I am almost sure that random forest will perform better than “nnet”. If you want to switch to a method without too many hyperparameters, why not random forest or gradient boosting tree?

Response: We thank Reviewer #2 for the comment.  We switched from “deepnet” to “nnet” [21] since one of the previous reviewers had suggested us to use a recent algorithm. The “nnet” is a just published (in 2020) neural network based deep learning method and more efficient for classification.  “deepnet” is quite older (developed in 2015), and it does not have option to compute classification metrices like True Positive (TP), False Negative (FN), False Positive (FP), and True Negative (TN). Thus, we were not able to evaluate its classification performance such as accuracy, sensitivity, precision, specificity, area under curve (AUC) through “deepnet”. On the other hand, using “nnet”, we can easily compute all classification metrices including accuracy and AUC.

Of note, we here used the default setting of “nnet” as follows: (i) size (= number of units in hidden layer) (= 2), (ii) rang (= initial random weights)(= 0.1) while [-rang, rang], (iii) decay (= parameter for weight decay)(= 5 x e-4), (iv) maxit (= the maximum number of iterations or number of epochs)(= 100), (v) MaxNWts (= the maximum allowable number of weights)(= 84581) and other default parameters.

We used this deep learning technique with 10-fold cross validation to examine the class-label (normal and Uterine Cervical cancer groups) of the differentially expressed genes with a repeat of 30 times. In each fold of test set, we ran “nnet” tool using maxit (number of epochs) equal to 100, that means the deep learning method was internally repeated for 100 times, and then computed the classification metrics at one time iteration of each fold. Thus, our deep learning method had already repeated 30,000 times (30 x 10 x 100) from which we computed average accuracy, where every sample has been used as test set at least once (i.e., no sample is missing as test sample). So, the average accuracy that we obtained was not by chance.  See pages 6-8, line 194-222 of Section 3.

Furthermore, we provided the link of our R code in our manuscript. In addition, we used user-defined “seed value” during cross-validation in R code.  So, entire classification result can be reproducible. See Abstract, page 1, lines 25-26.

Any deep learning/neural network method (including “nnet”) and Random Forest classifier serve totally different purposes. Thus, the direct comparative study between them is difficult, and it might be biased to some extent. In general, Random Forest and Gradient Boosting perform well to classify tabular, structured and small sized data with limited number of samples (e.g., the data consisting of 1,000 features and 100 samples). On the other hand, in real world problems, deep learning/neural network method is the preferred choice due to its big size and its variety to work in different kinds of data including unlabeled/unstructured data (e.g., text, images, audio and video). It is obvious that with a big data, deep learning is surely more powerful than any kind of method of different family. Another key to get success with a deep learning / neural network is to model the correct architecture. Whenever sample size is very large (>1k approximately) and data are sparse, deep learning / neural network is expected to perform better than random forest or gradient boosting tree method. However, deep learning method performs well for this case, but it might vary in performance by data. Normally it performs well with larger number of samples. Here, we used a medium sized methylation dataset having a limited number of samples (=215). However, currently there are many methylation data in large projects like TCGA and ICGC, where the number of samples is large (>1k) (including single cell methylation data that might have sparse data). Thus, to make our method more generalized prone to big data analysis, we applied a deep learning method in our framework.

To follow the suggestion of Reviewer #2,  in our future study, we will extend our current work through integrating random forest ensemble method into deep learning strategy to obtain a better classification model in all prospective, and then apply that on big data (e.g., single cell RNA sequencing data or other TCGA cancer tissue specific data) for cancer classification. We mentioned this in the end of Conclusion, pages 12-13, lines 299-302.